# An Archetypical Model for Engrafting *Bacteroides fragilis* into Conventional Mice Following Reproducible Antibiotic Conditioning of the Gut Microbiota

**DOI:** 10.3390/microorganisms11020451

**Published:** 2023-02-10

**Authors:** Osagie A. Eribo, Charissa C. Naidoo, Grant Theron, Gerhard Walzl, Nelita du Plessis, Novel N. Chegou

**Affiliations:** 1DSI-NRF Centre of Excellence for Biomedical Tuberculosis Research, South African Medical Research Council Centre for Tuberculosis Research, Division of Molecular Biology and Human Genetics, Department of Biomedical Sciences, Faculty of Medicine and Health Sciences, Stellenbosch University, Cape Town 8000, South Africa; 2African Microbiome Institute, Faculty of Medicine and Health Sciences, Stellenbosch University, Cape Town 8000, South Africa

**Keywords:** gut microbiota, metronidazole, *Bacteroides fragilis*, imipenem, clindamycin

## Abstract

*Bacteroides fragilis* is a commonly investigated commensal bacterium for its protective role in host diseases. Here, we aimed to develop a reproducible antibiotic-based model for conditioning the gut microbiota and engrafting *B. fragilis* into a conventional murine host. Initially, we selected different combinations of antibiotics, including metronidazole, imipenem, and clindamycin, and investigated their efficacy in depleting the mouse *Bacteroides* population. We performed 16S rRNA sequencing of DNA isolated from fecal samples at different time points. The α-diversity was similar in mice treated with metronidazole (MET) and differed only at weeks 1 (*p* = 0.001) and 3 (*p* = 0.009) during metronidazole/imipenem (MI) treatment. *Bacteroides* compositions, during the MET and MI exposures, were similar to the pre-antibiotic exposure states. Clindamycin supplementation added to MET or MI regimens eliminated the *Bacteroides* population. We next repeated metronidazole/clindamycin (MC) treatment in two additional independent experiments, followed by a *B. fragilis* transplant. MC consistently and reproducibly eliminated the *Bacteroides* population. The depleted *Bacteroides* did not recover in a convalescence period of six weeks post-MC treatment. Finally, *B. fragilis* was enriched for ten days following engraftment into *Bacteroides*-depleted mice. Our model has potential use in gut microbiota studies that selectively investigate *Bacteroides’* role in diseases of interest.

## 1. Introduction

The gut microbiota plays an essential role in shaping and modulating host physiological and immune processes at proximal and peripheral sites [1,2,3,4,5,6]. Our knowledge of the community dynamics and diversity of commensal microbes residing in the gut (gut microbiota) has increased in the last two decades, especially with the development of culture-independent techniques. It is estimated that the gut microbiota consists of roughly 100 trillion microbes, outnumbering host cells [7,8].

The gut microbiota is a highly dynamic ecosystem influenced by age, host genotype, and lifestyle [9]. Perturbations to the gut microbiota have been linked to an increased risk of inflammatory bowel disease, diabetes, obesity, colon cancer, and Parkinson’s disease [10,11,12,13,14,15,16,17,18,19,20,21,22,23]. Recent studies have also linked alterations in the gut microbiota composition to compromised host immune responses during *Mycobacterium tuberculosis* infection, leading to increased susceptibility [4,6,24,25]. As a result, there is a growing interest in understanding how the microbiota interacts with the host (as a whole or as discrete members), which may aid in developing microbiota-targeted therapeutics for several human diseases. Therefore, establishing animal models for testing with distinct microbiota is crucial for distinguishing causal from correlative effects in disease.

The major gut microbiota phyla include Bacteroidetes, Firmicutes, Proteobacteria, Actinobacteria, Fusobacteria, and Verrucomicrobia. The most abundant are the Firmicutes and Bacteroidetes [26,27]. *Bacteroides* belonging to the Bacteroidetes phylum constitute a substantial proportion of the microbiota composition and are predominantly anaerobic [28,29]. More than twenty species have been classified within the *Bacteroides* genus; however, *Bacteroides fragilis* is a commonly studied gut commensal for its potential immunomodulatory properties and protective role in different diseases [28,30,31,32,33,34].

Indeed, studies have shown that *B. fragilis,* together with its capsular polysaccharide-A molecule (PSA), can induce host pro-and anti-inflammatory processes and cytokines such as interferon (IFN)-γ, interleukin (IL)-10, and IFN-β [35,36,37]. Induction of host immune response and cytokines by *B. fragilis* occurs via antigen-presenting cell (APC)-major histocompatibility complex (MHCII) presentation of the processed fragment of PSA to CD4 T-cells similar to proteins, a characteristic unique to PSA as well as other zwitterionic polysaccharides [38,39]. Accordingly, many studies have reported the protective role of *B. fragilis* or PSA in diseases such as encephalomyelitis [40,41], pulmonary inflammation [36], viral infection [34,42], colitis [43], colorectal cancer [31,32], and asthma [44] in different mouse models.

Germ-free (GF) and antibiotic-treated conventionally colonized wild-type mice are valuable tools for investigating host-microbiota relationships. Generally, GF mice models are considered to be the gold standard in studying the contribution of single microbiota species during host disease. As an advantage, GF mice allow for the generation of gnotobiotic animals colonized solely by the target microbe of interest, enabling the investigation of specific gut microbiota-disease relationships in the complete absence of other gut microbes. However, GF mouse models are expensive and require specialized skills and facilities, limiting their use, especially in resource-constrained settings. Aside from the cost of maintaining GF animals, there are other inherent limitations to consider in their use. For example, the mucosal immune system of GF mice is underdeveloped, containing few immunoglobulin-A-secreting plasma cells, small lymphoid follicles, and reduced submucosal T-cell populations [45]. Therefore, specific microbiota-induced immune responses in these animals may be naturally defective or skewed. In addition, microbiota-initiated immune responses in GF mice may not precisely represent conventional systems, given that under normal circumstances, a host is never totally sterile. Thus, complementary conventional mouse models that leverage antibiotic manipulation of the gut microbiota are needed.

Whole or selective manipulation of the gut microbiota with antibiotics has remained a popular, cheap, and viable technique for studying the gut microbiota’s role in disease. Still, reproducibility in many antibiotic models is challenging and not unexpected, considering the variability in the choice of antibiotics, spectrum, duration of treatment, and the delivery method used in different studies. Here, we aimed to develop a reproducible antibiotic-based model for conditioning the gut microbiota and engrafting *B. fragilis* into a conventional wild-type murine host, which could be valuable for investigating the role of *Bacteroides* in diseases of interest.

## 2. Materials and Methods

### 2.1. Mice

Eight-week-old inbred female C57BL/6J mice, which are conventionally colonized by gut microbiota, were used in the study. The mice were fed an autoclaved diet and water and were housed under pathogen-free conditions in temperature- and humidity-controlled cages on a 12 h light-dark cycle in the Animal Biosafety Level 3 (ABSL3) Laboratory at Tygerberg Campus, Stellenbosch University. The experimental protocol was approved by the Stellenbosch University Animal Care and Use Ethics Committee (ACU-2019-8993 and ACU-2020-15458).

### 2.2. Antibiotic Treatment

We initially treated four different groups of mice (4–5 per group) separately with either metronidazole (5 mg/mL) or a combination of metronidazole (5 mg/mL) + clindamycin (10 mg/mL), metronidazole (5 mg/mL) + imipenem (5 mg/mL), or metronidazole (5 mg/mL) + imipenem (5 mg/mL) + clindamycin (10 mg/mL) (Sigma Aldrich, USA) (Figure 1A). All mice received a daily dose of 200 µL of antibiotic preparations via the oral gavage for four weeks. The metronidazole (5 mg/mL) + clindamycin (10 mg/mL) treatment was repeated in two additional independent experiments using different sets of mice (Figure 1B,C). Fresh antibiotics were prepared at every 3-day interval.

### 2.3. Bacteroides Fragilis Strain and Engraftment into Antibiotic-Treated Mice

We obtained *Bacteroides fragilis* NCTC 9343 from the National Collection of Type Cultures (London, UK). The bacterium was reactivated following the manufacturer’s protocol. *B. fragilis* was grown in Brain Heart Infusion (BHI) broth (Sigma Aldrich, St. Louis, MO, USA) and used for transplantation in the antibiotics-treated mice. The culture was incubated anaerobically at 37 °C for 24 h [34]. A 5 × 10^7^ CFU/mL culture suspension of *B. fragilis* was centrifuged at 3000× *g* for 15 min. Bacterial pellets were resuspended in phosphate-buffered saline and washed twice. Viability was determined by plating an aliquot of serially diluted bacterial suspension on blood agar (Sigma Aldrich, St. Louis, MO, USA). The mice were gavaged with 200 µL of *B. fragilis* suspension a day after stopping antibiotics (Figure 1C).

### 2.4. Stool Collection, DNA Isolation, and Quality Control

We collected approximately 200–250 mg of fresh stool pellets from each mouse in the different treatment groups. In the first experiment, the stool was collected at five time points: weeks 0, 1-, 3-, 4-, and two weeks post-antibiotic administration (Figure 1A). For the additional two experiments with metronidazole and clindamycin, the stool was collected at (i) week 4 during treatment and weeks 2-, 4-, and 6- post-antibiotics (Figure 1B) and (ii) week 3-post and day 10-post *B. fragilis* gavage, respectively (Figure 1C). DNA was isolated from all mice using the QIAamp PowerFecal Pro DNA Kit (Qiagen, Germantown, MD, USA), following the manufacturer’s protocol. Isolated DNA was quantified on the Qubit 4.0 Fluorometer using the Qubit 1× dsDNA HS assay kit (Thermo Fisher Scientific, Loughborough, UK). Spectrophotometry was thereafter performed on the NanoDrop^®^ ND-1000 to assess the purity of the DNA samples. Genomic quality scores (GQS) were determined on the LabChip GXII Touch using the DNA Extended Range LabChip and Genomic DNA Reagent Kit (PerkinElmer, Waltham, MA, USA), CLS140166 Rev.C (Supplementary Report: Genomic DNA (gDNA) Quality Control). The DNA was stored at −80 °C until amplification and sequencing.

### 2.5. 16S rRNA Gene Amplification

The Ion 16S™ Metagenomics Kit (Thermo Fisher Scientific, Loughborough, UK) was used to amplify seven hypervariable regions of the 16S rRNA gene (V2, V3, V4, V6, V7, V8, V9). Target regions were amplified from 2 µL of DNA across 25 cycles, with two primer pools on the SimpliAmp Thermal Cycler (Thermo Fisher Scientific, Loughborough, UK) following the protocol MAN0010799 REV C.0. The presence of amplified products was verified on the PerkinElmer LabChip^®^ GXII Touch (PerkinElmer, Waltham, MA, USA), using the X-mark chip and HT DNA NGS 3K reagent kit according to the manufacturer’s protocol CLS145098 Rev. E. PCR products from the two primer pools were combined for each sample, purified with Agencourt™ AMPure™ XP reagent (Beckman Coulter, Brea, CA, USA), and eluted in 15µL nuclease-free water. Purified amplicons were quantified on the Qubit 4.0 Fluorometer using the Qubit 1× dsDNA HS assay kit (Thermo Fisher Scientific, Loughborough, UK).

### 2.6. Library Preparation

Using the Ion Plus Fragment Library Kit, library preparation was performed from 100 ng pooled amplicons for each sample. Briefly, 79 µL of each purified, pooled PCR product was end-repaired at room temperature for 20 min, using 1 µL end-repair enzyme and 20 µL end-repair buffer in a final volume of 100 µL. The end-repaired products were purified with Agencourt™ AMPure™ XP reagent (Beckman Coulter, Brea, CA, USA) and ligated to 2 µL IonCode™ Barcode Adapters (Thermo Fisher Scientific, Loughborough, UK). The adapter-ligated, barcoded libraries were further purified with Agencourt™ AMPure™ XP reagent (Beckman Coulter, Brea, CA, USA) and quantified using the Ion Universal Library Quantitation Kit. qPCR amplification was performed using the StepOnePlus™ real-time PCR system (Thermo Fisher Scientific, Loughborough, UK), and library fragment size distributions were assessed on the LabChip^®^ GXII Touch (PerkinElmer, Waltham, MA, USA), using the X-mark chip and HT DNA NGS 3K reagent kit according to the manufacturer’s protocol.

### 2.7. Template Preparation, Enrichment, and Sequencing

Libraries were diluted to a target concentration of 10 pM. The diluted, barcoded 16S libraries were combined in equimolar amounts for template preparation using the Ion 510™, Ion 520™, and Ion 530™ Chef Kit (Thermo Fisher Scientific, Loughborough, UK). Briefly, 25 µL of the pooled library was loaded on the Ion Chef liquid handler, and enriched template positive ion sphere particles were loaded onto an Ion 530™ Chip (Thermo Fisher Scientific, Loughborough, UK). Massively parallel sequencing was performed on the Ion S5™ Prime using the Ion S5™ Sequencing Solutions and Sequencing Reagents Kits following the manufacturer’s protocol.

### 2.8. Data Analysis

Flow space calibration and BaseCaller analyses were initially performed using the default analysis parameters in the Ion Torrent Suite Version 5.12.0 software. Sequences were demultiplexed and quality filtered in QIIME 2 (version 2020.2) [46]. Fastq files were imported (in SingleEndFastqManifestPhred33V2 format) into QIIME 2 using *qiime tools import*. The DADA2 [47] plugin was used for denoising *(qiime dada2 denoise-pyro*) and clustering. Merged representative sequences were aligned to the Greengenes 13.8 reference sequence using the QIIME 2 fragment insertion method SATé-Enabled Phylogenetic Placement (SEPP) [48], and the resulting insertion tree was used for downstream analyses. Taxonomic classification was performed using the *q2-vsearch* plugin [49]. Sequences are available in the Sequence Read Archive (PRJNA820505). Downstream analyses were carried out in RStudio (v1.3.1) as previously described [50]. Briefly, α-diversity was calculated using Faith’s Phylogenetic Diversity (Faith’s PD) index, and β-diversity was computed using weighted UniFrac in *vegan*. Differential abundance testing was performed in *DESeq2* with Benjamini–Hochberg multiple comparison adjustments, where adjusted *p*-values < 0.2 were considered statistically significant.

### 2.9. Statistical Analysis

Statistical analyses were performed in GraphPad Prism version 8 (Graphpad Software Inc., San Diego, CA, USA). Differences in α-diversity were assessed with the Friedman’s test, followed by the Dunn’s multiple comparison post-hoc test. Permutational multivariate analysis of variance (PERMANOVA) testing was performed to compare β-diversity. *p*-values less than 0.05 were considered statistically significant unless otherwise specified.

## 3. Results

### 3.1. Clindamycin Supplementation Rapidly Reduces Gut Microbiota Diversity

An assessment of the activity of antimicrobials against *Bacteroides* shows low resistance to metronidazole and carbapenems (imipenem) (<5–10%), while clindamycin was moderately active, with 10 to 60% resistance in strains of *B. fragilis* group [51,52]. Infections with clinical strains of the *B. fragilis* are routinely treated with metronidazole [53]. However, how these commonly used antibiotics shape the *Bacteroides* population in a microbiota setting and the time it takes for recovery remain to be determined. We first determined the most effective combination of antibiotics to deplete the entire *Bacteroides* population.

The α-diversity did not change during metronidazole (MET) treatment (Figure 2A). In metronidazole/imipenem (MI)-treated mice, a reduction in α-diversity was observed at weeks 1 and 3, relative to the baseline (week 0) (Figure 2B). The most significant reduction in α-diversity was seen when clindamycin was added to the antibiotic combination (Figure 2C,D). β-diversity differed between baseline and subsequent time-points in each antibiotic combination (Figure 3A–D).

Bacteroidetes, Firmicutes, Actinobacteria, and Proteobacteria were the most abundant phyla at baseline (Figure 4A–D). Overall, MET- and MI-treated mice had similar microbial taxa composition between the antibiotics’ pre-treatment state and throughout treatment (Figure 4A,B). Clindamycin-containing combinations (MC and MIC) depleted mice of all phyla except Proteobacteria within the first week of treatment (Figure 4C,D). This effect persisted at weeks 3 and 4 in MC-treated mice (Figure 4C), whereas MIC-treated mice showed some recovery of Firmicutes from week 3 onwards (Figure 4D). *Bacteroides* population was essentially unchanged at the genus level in the MET and MI-treated mice (Figure 5A,B). *Prevotella* and *Lactobacillus* were mainly depleted, while *Sutterella* and *Bifidobacterium* were represented during MET and MI treatment, as visualized in Figure 5A,B.

*Bacteroides* did not recover in MC and MIC-treated mice two weeks post-treatment (Figure 5C,D). In MC-treated mice, *Sutterella* and *Helicobacter* were represented at week 4, while *Sutterella*, *Oscillospira*, *Helicobacter*, *Enterococcus,* and *Ruminococcus* were also seen two weeks post-treatment (Figure 5C). *Sutterella, Ruminococcus,* and *Helicobacter* were the most represented in the MIC group at week 4 during treatment, while *Bifidobacterium*, *Sutteralla, Lactobacillus, Oscillospira,* and *Ruminococcus* were present post-treatment (Figure 5D).

### 3.2. Bacteroides Did Not Recolonize the Gut following a Six-Week Convalescence Period

Given the efficacy of clindamycin-containing antibiotics in depleting *Bacteroides*, we repeated the metronidazole/clindamycin (MC) treatment in a different set of mice to determine the reproducibility of the results. We treated the mice with MC for four weeks via daily oral gavage and measured microbiome changes at the end of treatment. To determine the time point at which *Bacteroides* would begin to recover, we extended the post-treatment convalescence window to weeks 4 and 6. MC reduced α-diversity at week 4 and began to recover by week 2 post-treatment (Figure 6A). No differences were observed between baseline and weeks 4 and 6 post-treatment (Figure 6A). β-diversity, on the other hand, differed between the baseline and all subsequent time points (Figure 6B).

As with the first experiment, treating mice with MC for four weeks completely depleted all phyla except Proteobacteria. The phyla composition at weeks 2, 4, and 6 post-treatment was similar but clearly distinguishable from baseline. Bacteroidetes, Defferibacteres, and TM7 (Saccharibacteria phyla) remained completely depleted six weeks after stopping MC treatment. On the other hand, Firmicutes had recovered at W2-post MC treatment and, together with Proteobacteria, dominated at weeks 2, 4, and 6 post-treatment (Figure 6C).

As observed in the first experiment, *Sutterella* and *Helicobacter* were the most represented genera during MC-treatment (Figure 6D). As with phyla comparisons, genera at weeks 2, 4, and 6 post-treatment were similar but differed from baseline (Figure 6D). *Bacteroides* did not recover at week 6 post-MC treatment, whereas *Sutterella, Ruminococcus, Enterococcus, Oscillospira,* and *Helicobacter* were present at weeks 2, 4, and 6 after treatment was stopped (Figure 6D).

### 3.3. Engraftment of Bacteroides fragilis into MC-Treated Mice Reshapes the Gut Microbiota

Following the depletion of *Bacteroides*, we next aimed to engraft *B. fragilis* into MC-treated mice. MC treatment for three weeks depleted the α-diversity, and no difference was seen ten days following *B. fragilis* inoculation (Figure 7A). β- diversity was altered during MC treatment and after the *B. fragilis* gavage (Figure 7B).

Again, Proteobacteria predominated during treatment, with all other phyla completely depleted (Figure 7C). Proteobacteria, Bacteroidetes, and Firmicutes returned to baseline ten days after *B. fragilis* engraftment, while Actinobacteria, Deferribacteres, TM7, and Tenericutes did not recover (Figure 7C). *Bacteroides* were highly represented ten days following *B. fragilis* gavage (Figure 7D). Differential abundance analyses showed enrichment of *Aggregatibacter pneumotropica* and *Haemophilus parainfluenzae* and depletion of *Parabacteroides distasonis* and *Bacteroides acidifaciens* at week 3 versus week 0 (Figure 8A). Following *B. fragilis* gavage, *B. fragilis* and *B. ovatus* were enriched, and several taxa were depleted relative to week 0 (*Mucispirillum schaedleri*, *Parabacteroides distasonis*) (Figure 8B) and week 3 (*Aggregatibacter pneumotropica*, *Haemophilus parainfluenzae*) (Figure 8C).

## 4. Discussion

There is an increased interest in investigating the role of individual gut microbiota species in host disease processes. Polysaccharide producing-*Bacteroides fragilis* contribute to the host’s protection in some reported disease cases [30,31,32,42]. Based on the available data on the relationship between *Bacteroides* such as *B. fragilis* and host immunity, we could expect new studies to emerge exploring the potential role of *B. fragilis* in yet unexplored diseases. Our study aimed to develop a reproducible, antibiotic-based model for conditioning the gut microbiota and engrafting *B. fragilis* into a conventional murine host, which could be valuable for investigating its role in diseases of interest.

Overall, the composition of *Bacteroides* in mice treated with either metronidazole or a combination of metronidazole and imipenem was similar to the antibiotic’s pre-treatment state. In marked contrast, the addition of clindamycin to the regimes rapidly and consistently depleted *Bacteroides* composition and other microbiota phyla, except for Proteobacteria. In an independent experiment, members of the *Bacteroides* genus did not recover in a six-week convalescence period. Finally, we showed that *B. fragilis* engrafted into the *Bacteroides*-depleted mice was highly enriched ten days following oral gavage and dominated the microbiota composition.

In some studies, *B. fragilis* has been regarded as a potential next-generation probiotic. Several *Bacteroides* strains have also been reported to play immunomodulatory or protective functions in intestinal and extraintestinal diseases. Work by Wang and colleagues showed that *B. acidifaciens* ameliorated liver injury by reducing the apoptosis of liver cells [54]. Mice reconstituted with *B. acidifaciens* had reduced clusters of differentiation (CD)95/CD95 ligand signaling, which decreased the L-glutathione/glutathione liver ratio and made them more resistant to alcoholic liver injury [54]. In another study, *oral inoculation of B. vulgatus* abrogated changes in the proportion of regulatory T-cells (Tregs) in the mesenteric lymph node and colon cytokine mRNA expression following lipopolysaccharide (LPS)-induced intestinal injury in mice [55]. Similarly, *B. fragilis* protected against LPS-induced systemic inflammation in mice, an effect that was attributed to its ability to induce the production of short-chain fatty acids (SCFA) and IL-10 secretion [56].

Furthermore, in a study by Sofi and colleagues, an increase in the *Bacteroides population* was associated with reduced Graft-versus-host (GVHD) disease in recipient mice that received fecal transplantation [30]. Oral inoculation of *B. fragilis* significantly improved acute and chronic GVHD disease. The authors further suggested that the protective properties of *B. fragilis* were likely due to an increase in SCFA, IL-22, and Tregs, which reduced inflammatory cytokine secretion and enhanced the integrity of gut tight junctions [30].

Although there are several antibiotic conditioning protocols in the literature, most studies do not consider the influence of closely related species that may share similar characteristics when conducting microbiota studies to investigate the role of single species using conventional mice. For example, the ability of *B. fragilis* to induce host immune responses is a feature attributed mainly to its zwitterionic polysaccharide production. Nevertheless, polysaccharide synthesis is a unique feature of *Bacteroides* species. Many species within the genus produce structurally similar polysaccharides to those produced by *B. fragilis* [42,57,58,59]. A study by Stefan and colleagues showed that *B. fragilis* and polysaccharide-A (PSA) isolated from *B. fragilis* regulated constitutive levels of interferon beta (IFN-β) production, which was protective against viral infections [42]. The same study also demonstrated that IFN-β induction was a shared feature of *Bacteroides* species. Outer membrane vesicles isolated from several *Bacteroides* species, including *B. thetaiotaomicron, B. dorei, B. uniformis, B. ovatus, and B. vulgatus,* also induced IFN-β, suggesting that the production of lipooligo/polysaccharides may represent a broader mechanism by which *Bacteroides* induce the immune response [42]. Furthermore, *B. thetaiotaomicron* has also been shown to ameliorate allergic airway inflammation, similar to *B. fragilis* or PSA [36,44,60].

Our model would allow the mechanistic study of the impact of a gut environment devoid entirely of *Bacteroides* during several chronic, inflammatory, and infectious diseases and investigate the role of *B. fragilis* in the absence of most species within the *Bacteroides* genus in a conventional system. For instance, understanding the gut microbiota’s role in tuberculosis (TB) and identifying species that may be involved in the protection or worsening of disease outcomes has recently become a topic of interest. Some animal studies suggest that gut microbiota alterations increase *Mycobacterium tuberculosis* (*Mtb*) susceptibility [4,24,25]. Our recent work also showed that specific gut anaerobes were abundant in the stool of TB patients and correlated with the upregulation of immune pathways associated with pro-inflammatory responses [50]. We recently also published a review article that discussed the role of *B. fragilis* in the immune response to viral infections and therapy, highlighting critical lessons that could stimulate new research to examine its possible role during *Mtb* infection [61]. Indeed, findings from a recent study by Yang and colleagues revealed that *B. fragilis* directly regulated a long non-coding RNA (lncRNA-CGB) during *Mtb* infection [62]. The lncRNA profile differs between germ-free and microbiota-colonized mice and plays a role in modulating host-microbiota interactions and has been implicated in regulating the host response to diseases such as obesity and cancer [63,64]. A study by Yang and colleagues demonstrated that oral inoculation of *B. fragilis* enhanced anti-TB immunity by promoting the expression of lncRNA-CGB through epigenetic modulation of IFN-γ expression [62].

While our protocol was specific for *B. fragilis*, it could be adapted to other *Bacteroides* species of interest. The concept may also be valuable in studies investigating species in other genera. We acknowledge the following limitations in our study. Although our protocol followed a longitudinal approach, where the baseline (week 0) microbiota composition was used to measure the effectiveness of the antibiotic treatment in each group, we did not include a naïve mice control group in order to eliminate possible background influences in the result. We did not also determine at what point, beyond six weeks, the *Bacteroides* population will begin to recover after stopping MC treatment. Furthermore, while we successfully established reproducibility in the dose of the MC combination used, we did not investigate the effect of different antibiotic concentrations. However, when deciding on the antibiotic combinations and doses, we considered the various concentrations and activities used in the available literature and their solubility (clindamycin is very soluble in water at 50 mg/mL, while metronidazole and imipenem are soluble at 10 mg/mL, respectively) and selected intermediate doses. As was shown in the result, MI combinations at concentrations of 5 mg/mL, respectively did not eliminate all members of the *Bacteroides* genus. Nevertheless, future studies following our concept could vary the concentrations of the antibiotics, especially clindamycin, in the mix to minimize the effect on the other microbiota as much as possible while retaining the impact on *Bacteroides*.

## 5. Conclusions

To summarize, we developed a new reproducible antibiotic-based model for conditioning the *Bacteroides* population and transplanting a single species into a conventional wild-type murine host. Additional studies may be needed to examine the long-term impact of metronidazole/clindamycin treatment beyond the time points investigated in our study. Nevertheless, our model already has the potential to be used in many microbiota studies, selectively investigating the broad role of *Bacteroides* in several diseases of interest or studies targeting a specific member of the genus, such as *B. fragilis.*

## Figures and Tables

**Figure 1 microorganisms-11-00451-f001:**
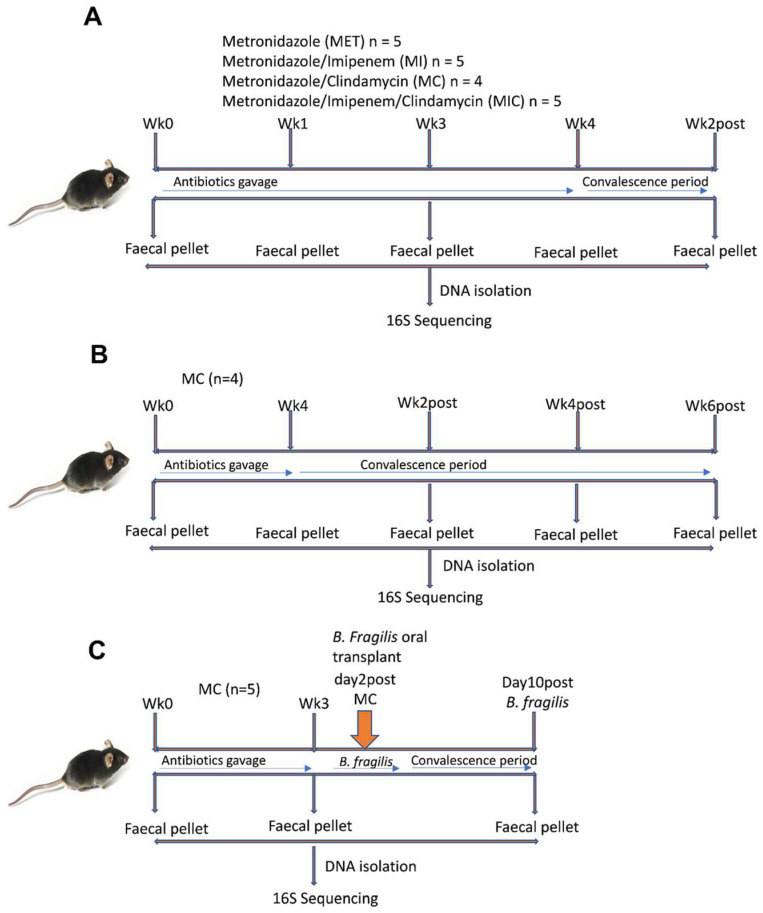
Experimental design for the different antibiotics’ treatment and B. fragilis engraftment (**A**) first experiment with four different groups of mice treated with metronidazole, metronidazole + imipenem, metronidazole + clindamycin, and metronidazole + imipenem + clindamycin, respectively (**B**) metronidazole + clindamycin repeated treatment in an independent experiment with convalescence timepoint extended to six weeks (**C**) metronidazole + clindamycin treatment followed by Bacteroides fragilis inoculation. n = 4–5 mice/group in each experiment.

**Figure 2 microorganisms-11-00451-f002:**
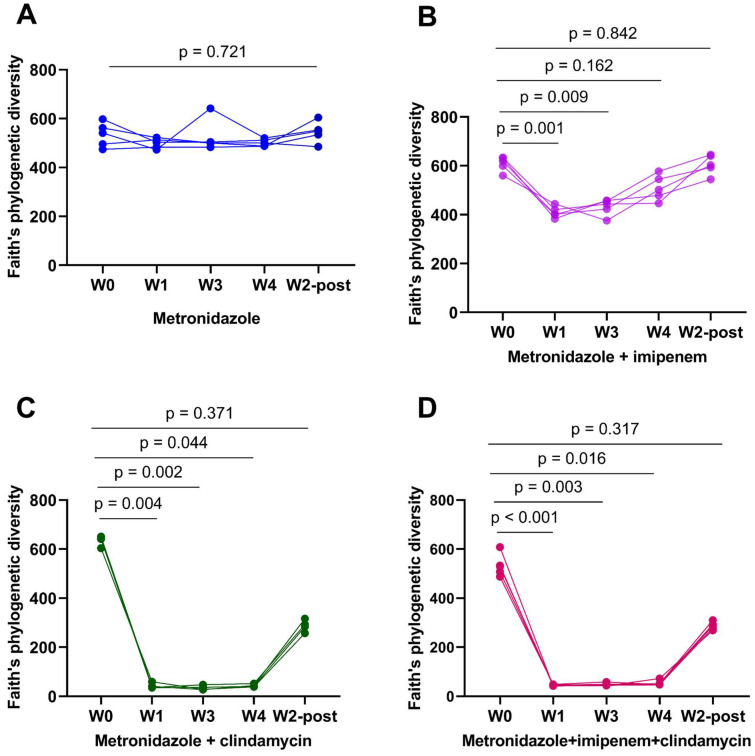
The addition of clindamycin rapidly reduces gut microbial diversity (**A**–**D**) alpha-diversity measured using Faiths’ phylogenetic index in metronidazole, metronidazole + imipenem, metronidazole + clindamycin, and metronidazole + imipenem + clindamycin-treated mice, respectively. *p* values for comparison between the baseline and the different time points are shown.

**Figure 3 microorganisms-11-00451-f003:**
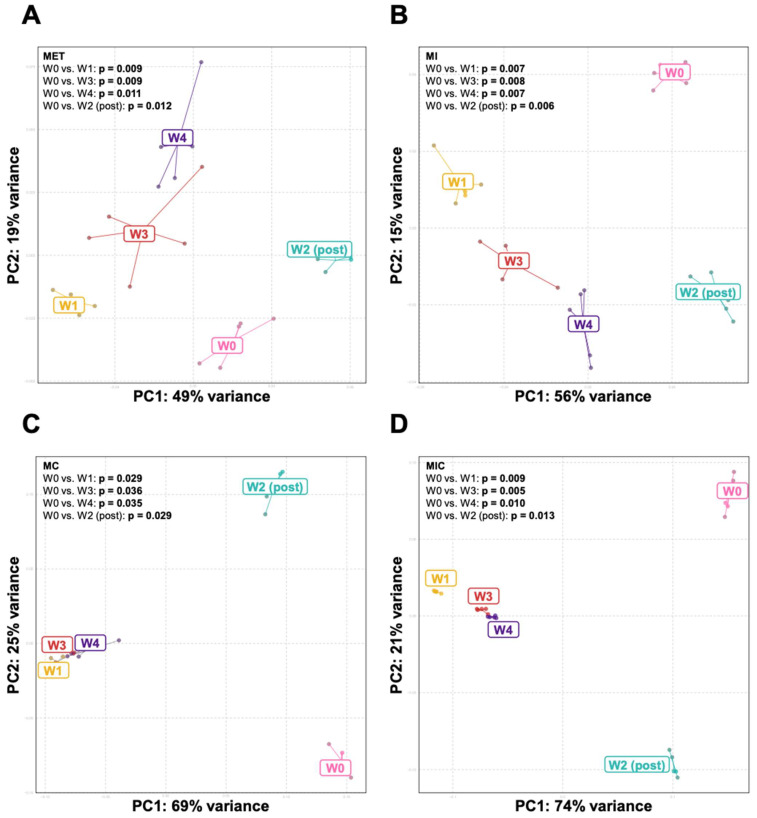
Principal coordinate analysis plot of weighted Unifrac distances in mice treated with (**A**) metronidazole, (**B**) metronidazole + imipenem, (**C**) metronidazole + clindamycin, and (**D**) metronidazole + imipenem + clindamycin. *p*-values are based on PERMANOVA analysis. n = 4–5 mice/group. PC: principal coordinate.

**Figure 4 microorganisms-11-00451-f004:**
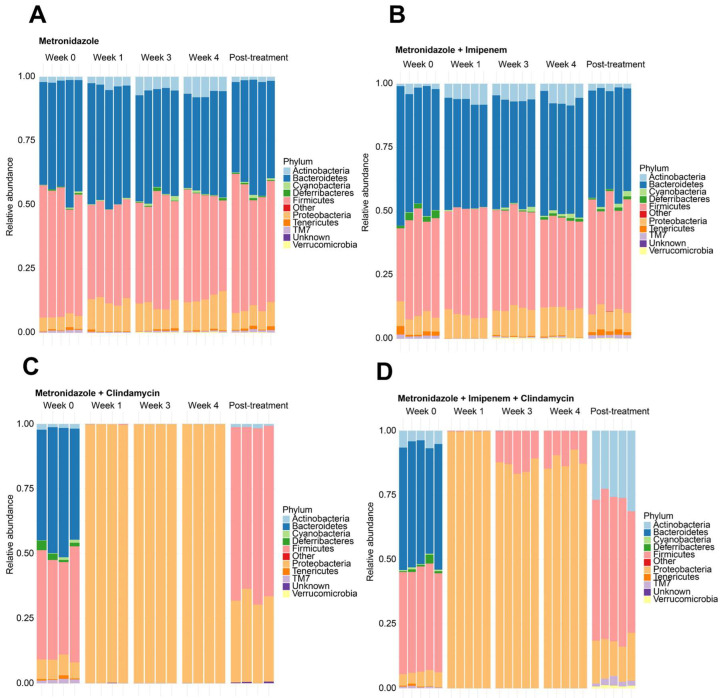
Stacked relative abundance plots at phyla level for mice treated with (**A**) metronidazole, (**B**) metronidazole + imipenem, (**C**) metronidazole + clindamycin, and (**D**) metronidazole + imipenem + clindamycin.

**Figure 5 microorganisms-11-00451-f005:**
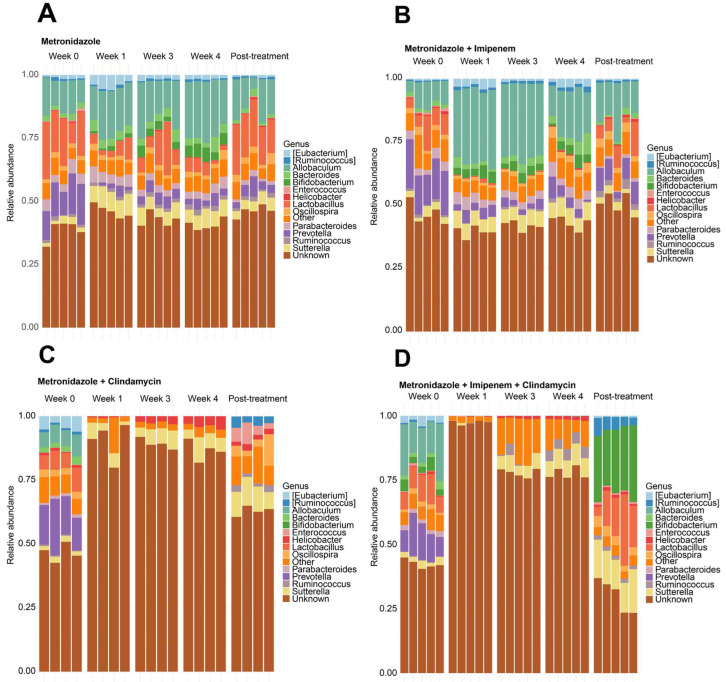
Stacked relative abundance plots at genus level for mice treated with (**A**) metronidazole (**B**) metronidazole + imipenem (**C**) metronidazole + clindamycin (**D**) metronidazole + imipenem + clindamycin.

**Figure 6 microorganisms-11-00451-f006:**
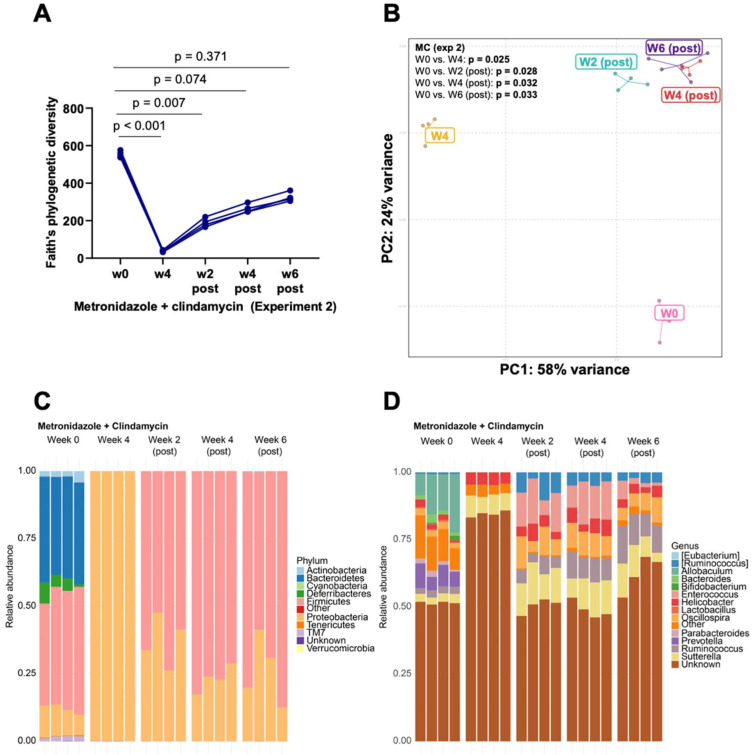
An independent experiment repeated with metronidazole + clindamycin treatment produces a reproducible result. *Bacteroides* did not repopulate the gut in a six-week convalescence period (**A**) α-diversity measured using Faith’s phylogenetic index. *p* values for comparison between the baseline and the different time-points are shown. n = 4. (**B**) Principal Coordinate Analysis plot of weighted Unifrac distances in the microbiota sequences in the mice treated with metronidazole + clindamycin (**C**) Stacked relative abundance plots at the phyla level (**D**) Stacked relative abundance plots showing microbiota differences at the genus level.

**Figure 7 microorganisms-11-00451-f007:**
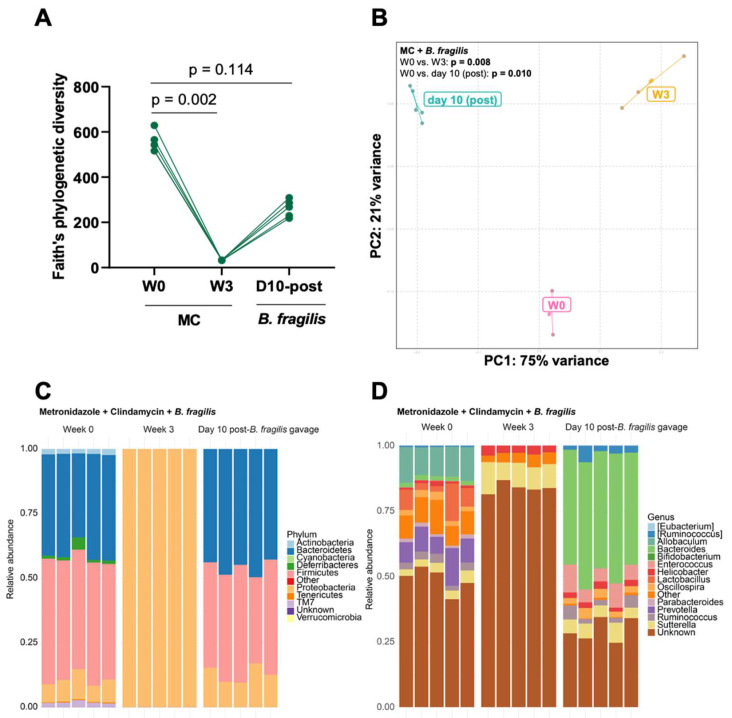
Transplantation of *B. fragilis* into MC-treated mice. (**A**) α-diversity measured using Faiths’ phylogenetic index. *p* values for comparison between the baseline and the different time points are shown. (**B**) Principal coordinate analysis plots illustrating differences in β-diversity based on weighted Unifrac. stacked relative abundance plots at the (**C**) genus level and (**D**) phylum level. (n = 5 mice/group).

**Figure 8 microorganisms-11-00451-f008:**
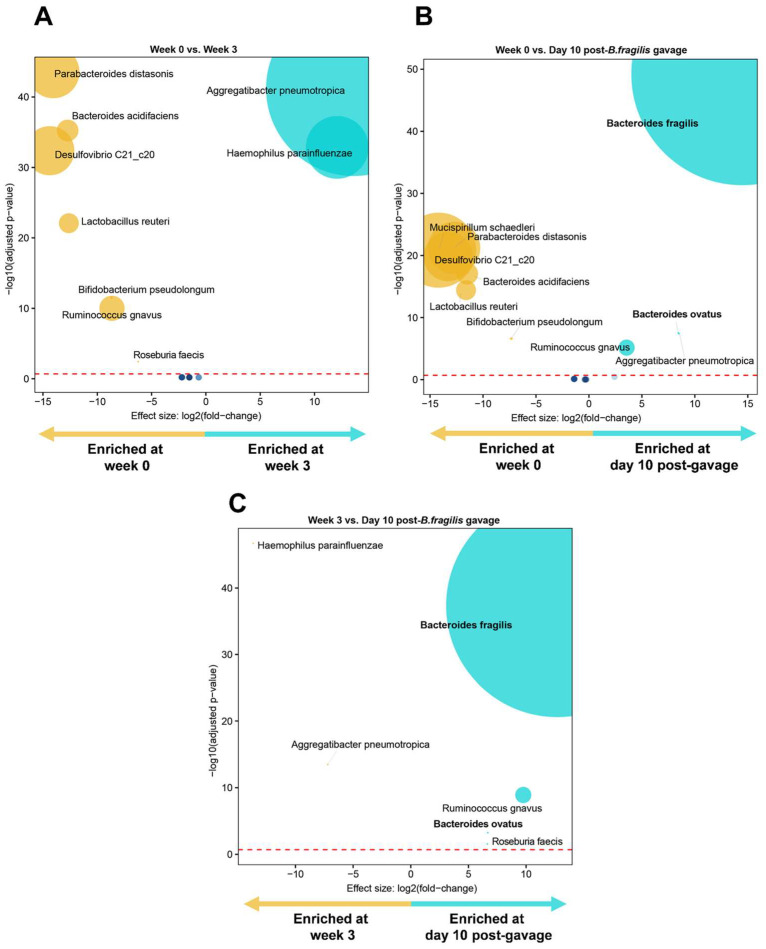
Volcano plots depicting differentially abundant taxa following transplantation of *B. fragilis* into MC-treated mice as determined by DESeq2 at (**A**) week 0 versus week 3, (**B**) week 0 vs. day 10 post-treatment, and (**C**) week 3 versus day 10 post-treatment. Significantly more discriminatory taxa appear closer to the left or right and are above the threshold (red dotted line, false discovery rate = 0.2). Relative abundance of taxa is indicated by circle size. *Bacteroides* spp. is bolded.

## Data Availability

Data sequence information can be accessed in the NCBI’s Sequence Read Archive (PRJNA820505): https://www.ncbi.nlm.nih.gov/sra (accessed on 6 December 2022).

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
