# Peer review of "An Archetypical Model for Engrafting Bacteroides fragilis into Conventional Mice Following Reproducible Antibiotic Conditioning of the Gut Microbiota"

_microorganisms, 2023, doi:10.3390/microorganisms11020451_

Round 1

Reviewer 1 Report

The authors have conducted an interestingly, well-structured and timely research, about the role of Bacteroides fragilis-animal models induced-antibiotic conditioning of the gut microbiota.

Minor recommendation:

At “2.9. Statistical analysis”, the p-value that was considered statistically significant should be inserted;

At figure 3. I suggest to include the meaning for PC1 (and 2) abbreviation; also, the authors can in text include name for TM7= Saccharibacteria phylum .

Author Response

Please see the responses to the comments below:

Comment 1: At “2.9. Statistical analysis”, the p-value that was considered statistically significant should be inserted.

Response 1: Thank you for highlighting this. We have inserted the below text where appropriate:

Lines 189-190: P-values less than 0.05 were considered statistically significant unless otherwise specified

Comment 2: At figure 3. I suggest to include the meaning for PC1 (and 2) abbreviation; also, the authors can in text include name for TM7= Saccharibacteria phylum .

Response 2: Thank you for the suggestion.

The meaning of the abbreviation “PC” has been added to the Figure 3 legend on line 216. “Saccharibacteria phyla” was added after TM7 on line 256.

Thank you for taking the time to review our manuscript.

Reviewer 2 Report

The manuscript written by Osagie A Eribo and colleagues is titled "An archetypical model for engrafting Bacteroides fragilis into conventional mice following reproducible antibiotic conditioning of the gut microbiota." Bacteroides fragilis is widely studied for its protective properties against host diseases as a commensal bacterium. The objective of this study was to develop a reproducible antibiotic-based model for conditioning the gut microbiota and engrafting B. fragilis into conventional murine hosts. Initially, we investigated the efficacy of different antibiotic combinations, including metronidazole, imipenem, and clindamycin, in depleting the mouse Bacteroides population. DNA isolated from faecal samples was sequenced at various time points using the 16S rRNA gene. α-diversity in mice treated with metronidazole (MET) was similar and differed only at weeks 1 and 3 during treatment with metronidazole/imipenem (MI). The composition of Bacteroides during MET and MI exposure was similar to that prior to antibiotic exposure. In addition to MET or MI regimens, clindamycin supplementation eliminated the Bacteroides population. Two additional independent experiments were conducted with metronidazole/clindamycin (MC), followed by a transplant of B. fragilis. MC consistently and reproducibly eliminated the Bacteroides population. The depleted Bacteroides did not recover during a convalescence period of six weeks post-MC treatment. Finally, B. fragilis was enriched ten days following engraftment into Bacteroides-depleted mice. I would like to make a few comments.

-Thanks to the authors for providing an informative introduction to the state of the art.

-How were the antibiotics selected? In addition, how was the concentration selected?

-What is the benefit of using Bacteroides fragilis NCTC 9343?

-It is recommended that Sections 2.8 and 2.9 be revised with respect to the information provided. The authors used 7 hypervariable regions. If other authors would like to repeat the methodology, they should be provided with instructions regarding the initial fastq.gz file to the final .txt file.

-Perhaps Figure 3 should be revised in terms of resolution

-I enjoyed reading youdocument. Thank you for providing this type of information.

Author Response

Please see the responses to the comments below:

Comment 1: How were the antibiotics selected? In addition, how was the concentration selected?

Response 1: Antibiotics were selected based on their antimicrobial activity against Bacteroides and other anaerobic bacteria. Furthermore, when deciding on the doses and combinations of the antibiotics, we considered both their solubility and the range of concentrations commonly used in other similar studies and selected intermediate doses for our treatment protocol. We discussed the considerations of antibiotics and doses used in our protocol in lines  383-389.

Comment 2: What is the benefit of using Bacteroides fragilis NCTC 9343?

Response 2: Following the elimination of Bacteroides in our model, we aimed to reconstitute the mouse microbiota with Bacteroides fragilis. For this, we obtained already isolated, purified and typed B. fragilis strain (B.fragilis strain NCTC 9343) from the National Collection of Type Cultures (London, England). As discussed in the manuscript, B. fragilis is a commonly studied commensal bacteria because of its immunomodulatory and protective benefits for various diseases (PMID: 33212011., PMID: 31089128., PMID: 35343370)

Comment 3: It is recommended that Sections 2.8 and 2.9 be revised with respect to the information provided. The authors used 7 hypervariable regions. If other authors would like to repeat the methodology, they should be provided with instructions regarding the initial fastq.gz file to the final .txt file.

Response 3: Thank you for this recommendation. We have updated Section 2.8 (lines 170-184) as below:

Data analysis

Flow space calibration and BaseCaller analyses were initially performed using default analysis parameters in the Ion Torrent Suite Version 5.12.0 Software. Sequences were demultiplexed and quality filtered in QIIME 2 (version 2020.2) [46]. Fastq files were imported (in SingleEndFastqManifestPhred33V2 format) into QIIME 2 using qiime tools import. The DADA2 [47] plugin was used for denoising (qiime dada2 de-noise-pyro) and clustering. Merged representative sequences were aligned to the Greengenes 13.8 reference sequence using the QIIME 2 fragment insertion method SATé-Enabled Phylogenetic Placement (SEPP) [48], and the resulting insertion tree used for downstream analyses. Taxonomic classification was done using the q2-vsearch plugin [49]. Sequences are available in the Sequence Read Archive (PRJNA820505). Downstream analyses were carried out in RStudio (v1.3.1) as previously described [50]. Briefly, α-diversity was calculated using Faith’s Phylogenetic Diversity (Faith’s PD) index, and β-diversity was computed using weighted UniFrac in vegan. Differential abundance testing was done in DESeq2 with Benjamini–Hochberg multiple comparison adjustments, where adjusted p-values < 0.2 were considered statistically significant.

Comment 4: Perhaps Figure 3 should be revised in terms of resolution

Response 4: We agree. Figures 3, 6 and 7 have been updated to reflect better resolution and colours (specifically for PCoA plots).

Thank you for reviewing our manuscript.